# Activation of the tick Toll pathway to control infection of *Ixodes ricinus* by the apicomplexan parasite *Babesia microti*

Marie Jalovecka[1,2], Laurence Malandrin[3], Veronika Urbanova[1], Sazzad Mahmood[1,2¤a], Pavla Snebergerova[1,2], Miriama Peklanska[1,2], Veronika Pavlasova[1,2¤b], Radek Sima[1], Petr Kopacek[1], Jan Perner[1], Ondrej Hajdusek[1] *

**1** Institute of Parasitology, Biology Centre, Czech Academy of Sciences, Ceske Budejovice, Czech Republic, **2** Faculty of Science, University of South Bohemia, Ceske Budejovice, Czech Republic, **3** Nantes-Atlantic National College of Veterinary Medicine (ONIRIS), UMR1300 BiOEPAR, National Research Institute for Agriculture, Food and the Environment (INRAE), Nantes, France

¤a Current address: Laboratory of Malaria and Vector Research, National Institute of Allergy and Infectious Diseases, Hamilton, Montana, United States of America
¤b Current address: School of Natural and Environmental Sciences, Newcastle University, Newcastle upon Tyne, United Kingdom
* hajdus@paru.cas.cz

## Abstract

The vector competence of blood-feeding arthropods is influenced by the interaction between pathogens and the immune system of the vector. The Toll and IMD (immune deficiency) signaling pathways play a key role in the regulation of innate immunity in both the *Drosophila* model and blood-feeding insects. However, in ticks (chelicerates), immune determination for pathogen acquisition and transmission has not yet been fully explored. Here, we have mapped homologs of insect Toll and IMD pathways in the European tick *Ixodes ricinus*, an important vector of human and animal diseases. We show that most genes of the Toll pathway are well conserved, whereas the IMD pathway has been greatly reduced. We therefore investigated the functions of the individual components of the tick Toll pathway and found that, unlike in *Drosophila*, it was specifically activated by Gram-negative bacteria. The activation of pathway induced the expression of *defensin* (*defIR*), the first identified downstream effector gene of the tick Toll pathway. *Borrelia*, an atypical bacterium and causative agent of Lyme borreliosis, bypassed Toll-mediated recognition in *I. ricinus* and also resisted systemic effector molecules when the Toll pathway was activated by silencing its repressor *cactus* via RNA interference. *Babesia*, an apicomplexan parasite, also avoided Toll-mediated recognition. Strikingly, unlike *Borrelia*, the number of *Babesia* parasites reaching the salivary glands during tick infection was significantly reduced by knocking down *cactus*. The simultaneous silencing of *cactus* and *dorsal* resulted in greater infections and underscored the importance of tick immunity in regulating parasite infections in these important disease vectors.

**Data Availability Statement:** All relevant data are in the manuscript and its supporting information files.

**Funding:** This work was mainly supported by the Czech Science Foundation (no. 20-05736S to PK). It was also supported by the Centre for Research of Pathogenicity and Virulence of Parasites (no. CZ.02.1.01/0.0/0.0/16_019/0000759), funded by the European Regional Development Fund (ERDF) and the Ministry of Education, Youth and Sport, Czech Republic (MEYS). MJ was supported by the Czech Science Foundation (no. 21-11299S). JP was supported by the Czech Science Foundation (no. 22-18424M). RS was supported by the Czech Science Foundation (no. 22-30920S). LM was supported by National Research Institute for Agriculture, Food and the Environment (INRAE, France). The funders had no role in study design, data collection and analysis, decision to publish, or preparation of the manuscript.

**Competing interests:** The authors have declared that no competing interests exist.

## Author summary

Ticks are important parasites of humans and animals worldwide. They act as vectors of numerous serious diseases such as Lyme disease, tick-borne encephalitis, rickettsiosis, babesiosis, and theileriosis. The innate immune system of ticks plays a crucial role in determining their vector competence and thus regulates the spread of pathogens. Through a series of RNA interference experiments in the European tick *Ixodes ricinus*, we were able to demonstrate the functionality of the tick Toll pathway and uncover its crucial role in both embryonic development and immunity. By identifying the read-out gene of the tick Toll pathway, *defensin*, we were able to show that the pathway can be stimulated by *Escherichia coli* and plays a key role in the balancing of *Babesia* infections. Our results provide insights into the functioning and evolutionary constraints of immune signaling pathways and contribute to the development of new defense mechanisms against arthropod-borne infections.

## Introduction

Tick-borne pathogens pose an ever-growing global threat. They cause serious diseases in humans and animals, including tick-borne encephalitis, Lyme disease (borreliosis), anaplasmosis, babesiosis, and theileriosis. The pathogens, which are ingested by the tick during a blood meal on an infected host, colonize the tick tissue, survive molting, and are transmitted to a new host throughout the next feeding. In order to spread in the tick body, the pathogens had to evolve mechanisms avoiding the tick immune defenses [1,2]. Innate immunity of the tick is a sophisticated system consisting of orchestrated humoral and cellular immune responses [1,3]. Humoral immunity is mediated by soluble immune peptides causing direct lysis of pathogens. Cellular immunity is then mediated by hemocytes that use pathogen-opsonizing molecules such as fibrinogen-related lectins [4] or thioester-containing proteins of the primordial complement system [5] to label and phagocytose the invading pathogens. The expression of immune peptides in response to the presence of microbes is a landmark of innate immunity in invertebrates. Infection is sensed and communicated at the molecular level primarily through the NF-κB (nuclear factor kappa-light-chain-enhancer of activated B cells) signaling pathways, Toll and IMD (immune deficiency) [6].

In *Drosophila*, the prompt NF-κB-mediated immune response is based on the recognition of structurally conserved molecules derived from microbes (peptidoglycans and β-glucans) [7]. The Toll pathway is triggered upon infection by Gram-positive bacteria or fungi [8]. Activation of the membrane-bound Toll receptor by the proteolytically cleaved protein Spätzle leads to the formation of the receptor-adaptor complex (including MyD88, Tube, and Pelle) and subsequent phosphorylation and degradation of the inhibitor of κB (I-κB) Cactus, which prevents nuclear translocation of the NF-κB transcription factor Dorsal by binding it and retaining it in the cytoplasm. In the nucleus, Dorsal binds to NF-κB sites in the promoter regions of effector genes and thus regulates their expression. In contrast, Gram-negative bacteria in *Drosophila* are recognized via the IMD pathway by activation of the peptidoglycan recognition protein PGRP-LC/LE. This leads to the formation of a receptor-adaptor complex (including IMD, FADD, and DREDD). As a result, the activated IKK (IkappaB kinase) complex phosphorylates the NF-κB transcription factor Relish, leading to its cleavage and dissociation of the ankyrin domains (analogous to the ankyrin domains of Cactus). Similar to Dorsal, the N-terminal part of the cleaved Relish then migrates into the nucleus and elicits the expression of its effector genes. Interestingly, both the Toll and IMD pathways have a significant

effect on the intensity of *Plasmodium* infection of blood-sucking mosquitoes [9], raising the question of whether ticks also use these pathways to control infections with tick-borne pathogens.

In this work, we have mapped and functionally characterized the Toll signaling pathway of the tick *Ixodes ricinus*, the main vector of human Lyme borreliosis in Europe. Although components of the Toll and IMD pathways have already been identified in ticks [10–12] and several members of the putative tick IMD pathway are involved in infection of ticks with *Borrelia* and *Anaplasma* [13–16], the tick Toll pathway has not yet been functionally characterized. We found that members of the Toll pathway are well conserved in *I. ricinus* and that this pathway influences both tick immunity and embryonic development. By identifying a specific read-out gene (*defensin*) of the tick Toll signaling pathway, we were able to show that the pathway can be strongly activated by *E. coli*, in contrast to *Drosophila*. In addition, stimulation or inhibition of the Toll pathway by RNA interference (RNAi) regulates the extent of tick infection with the *Plasmodium*-like pathogen *Babesia microti*.

## Results

### Components of the Toll pathway, but not the IMD pathway, are well conserved in ticks

To identify putative members of the NF-κB pathways in *I. ricinus* ticks, we searched our recently assembled transcriptome of tick nymphs [17]. We identified 9 of 13 (69%) and 8 of 17 (47%) putative components of the Toll and IMD pathways, respectively (S1 Fig and S1 Data). This result (summarized in Fig 1A) is consistent with previous observations in *Ixodes scapularis* and other chelicerates [10,11]. Among the core components of the pathway, the tick NF-κB transcription factor Dorsal is a typical protein of the classical *Drosophila* Toll pathway. It contains the Rel homology domain (RHD; dimerization domain), the Immunoglobulin-like fold, Plexins, Transcription factors (IPT; DNA-binding domain), the Nuclear localization signal (NLS), and the P/E/S/T-rich sequence (PEST) (Fig 1B). Although the tick NF-κB transcription factor Relish, a putative member of the IMD pathway in *Drosophila*, also contains all of these domains and phylogenetically aligns with Relish proteins from other arthropods (Fig 1C), it lacks the C-terminal sequence encoding ankyrin (ANK) domains that are critical for stabilization of Relish in the cytoplasm and its activation by the IMD pathway [18]. The truncated form of Relish was also the only form we identified in the most recent update of the *I. scapularis* genome (ASM1692078v2) and in transcriptomic bioprojects of several *Ixodes* ticks available in the NCBI database at the time of manuscript submission. In our transcriptome, we also identified two different homologs of NF-κB inhibitors (I-κB, Cactus): a classical I-κB Cactus1 (Cac1) and an atypical Cactus2 (Cac2), which has an NLS instead of the PEST sequence (Fig 1B). In addition, the tick transcriptome encodes all proteins of the Toll pathway that are required for the recognition and transmission of signals from the cell membrane to the nucleus. This is in contrast to the IMD pathway, which is incomplete and lacks crucial components such as PGRP-LC/LE and IMD.

To determine in which tissues the NF-κB pathway genes are expressed in ticks, we performed qRT-PCR expression profiling using tissues from feeding adult females and from different developmental stages before and after blood feeding. We observed dominant expression of *dorsal*, *relish*, and *cac1* in the ovaries and hemocytes (Fig 1D). *cac2* was mainly expressed in the hemocytes. As for developmental stages, these transcripts were mainly present in laid eggs (developing embryos) and unfed larvae. Based on these results, we hypothesized that the Toll signaling pathway in the tick may be involved in both development and immunity, similar to *Drosophila* [18,19].

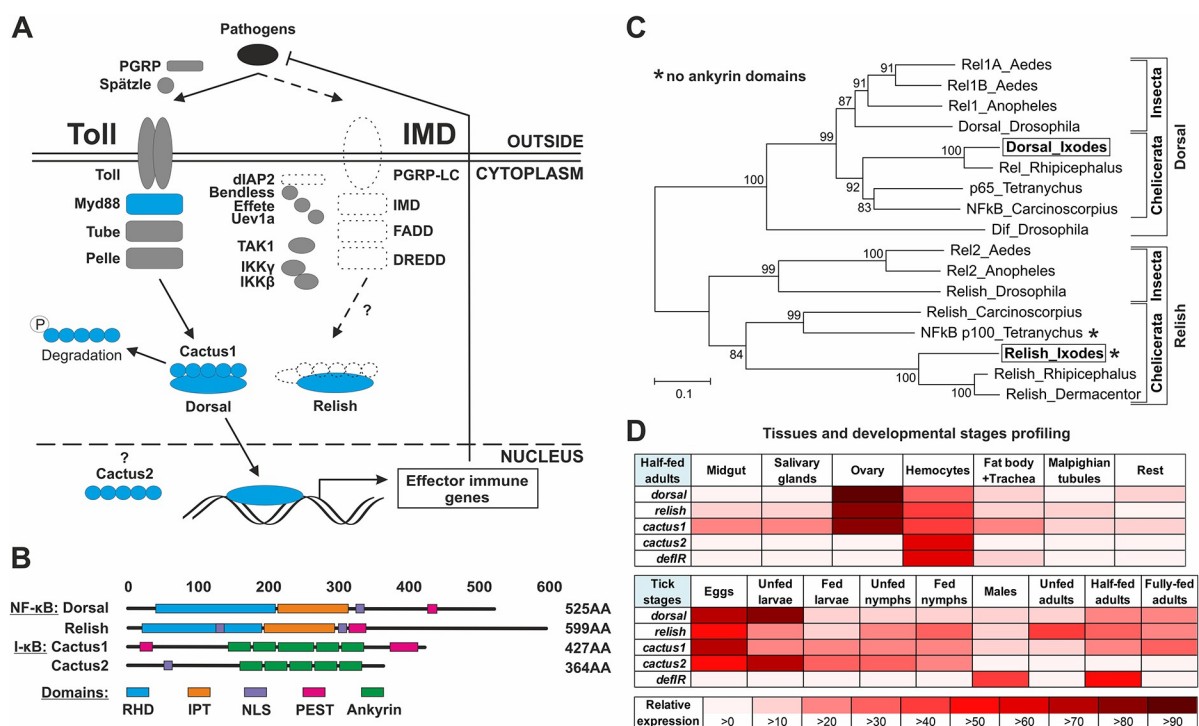

**Fig 1. The tick *I. ricinus* possesses a complete Toll pathway.** (A) Simplified schematic of the *I. ricinus* NF-κB signaling pathways (Toll and IMD). Proteins characterized in this work are highlighted in blue. Dashed lines indicate missing key components. P = phosphorylation. (B) Domain structure of two NF-κB and two I-κB proteins identified in *I. ricinus* transcripts. RHD = Rel homology domain, IPT = Immunoglobulin-like fold, Plexins, Transcription factors, NLS = nuclear localization signal, PEST = P/E/S/T-rich sequence, a signal peptide for protein degradation. (C) Phylogenetic tree of selected arthropod NF-κBs. Unrooted tree of NF-κB amino acid sequences reconstructed by the Neighbor-Joining method (NJ) based on alignment across the RHD domain with ClustalX. Asterisks indicate Relish sequences lacking ankyrin domains. Alignment and sequence descriptions can be found in the S2 Data and S1 Text. Numbers at branches represent bootstrap support using NJ with 1,000 repeats each (bars = 0.1 substitutions per site). (D) Relative expression (qRT-PCR) of tick *nf-κb*, *i-κb*, and *defensin* (*defIR*; tick antimicrobial protein; further identified as a Toll pathway read-out gene) in tissues of half-fed (fed for five days) adult ticks (top) and different developmental stages (bottom) normalized to tick *elongation factor 1* (*ef*). Results represent the mean of three independent biological replicates. The highest individual value for a given gene in each panel was set to 100% and all other values were expressed relative to this value.

## Dorsal is involved in tick embryogenesis and immunity

To test for involvement of the NF-κB transcription factor Dorsal in Toll signaling during tick development, we initially examined effects of *dorsal* depletion on embryogenesis, a process known in various arthropods to be dependent on Toll signaling [20]. Knockdown (KD) of *dorsal* by RNAi in feeding adult female ticks substantially reduced its transcript levels in the ovaries as well as in the laid eggs (S2 Fig). The duration of feeding and the weight of the ticks after feeding were not changed between the KD group and the control group. However, six weeks after oviposition, 80% of the eggs laid by the *dorsal* KD females failed to develop into embryos, whereas the majority (91%) of the embryos from control females injected with dsGFP developed normally. Larval hatching (total weight of all larvae from each egg clutch) was also significantly reduced in the *dorsal* KD group, although the effect was not as dramatic as during embryogenesis. This suggests that embryogenesis was delayed rather than blocked, probably due to insufficient suppression of *dorsal* transcript by the RNAi method. Together, these data indicate that depletion of the transcription factor Dorsal, a putative component of the tick Toll pathway, affects tick embryogenesis.

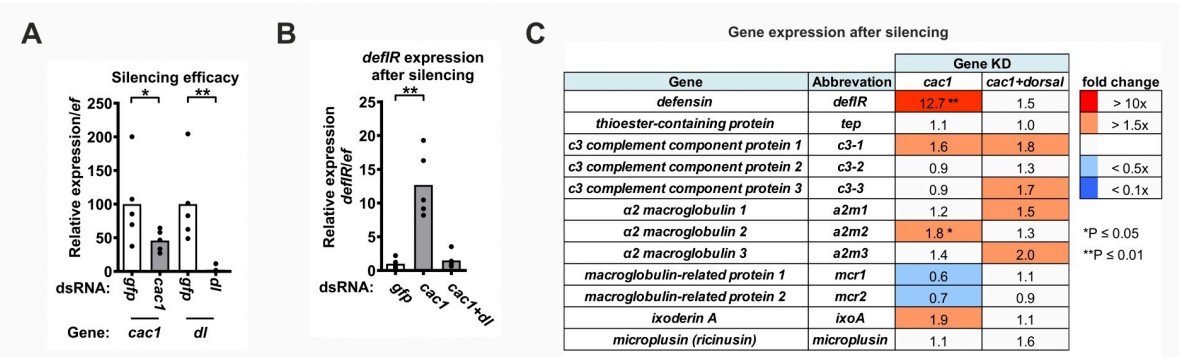

**Fig 2. The tick Toll pathway components regulate expression of immune genes.** (A) Efficacy of gene silencing in the fully-fed nymphs (whole bodies) measured by qRT-PCR. Each dot represents a pool of five nymphs. (B-C) Identification of the Toll pathway read-out effector gene. Relative expression (qRT-PCR) of *defIR* (B) and other immune genes (C) in the dsRNA-injected, fully-fed nymphs. Gene expression in the dsGFP control was set as 1. *dl* = *dorsal*, KD = knockdown. *P ≤ 0.05; **P ≤ 0.01.

To investigate the involvement of the Toll pathway in the tick immune defense, we used RNAi-mediated silencing of I-κB *cactus*, which typically leads to overexpression of effector genes regulated by the corresponding transcription factor Dorsal. When *cactus* is silenced together with *dorsal*, the expression of effector genes should remain at the unstimulated level. We focused exclusively on Cac1 because although Cac2 contained the ankyrin repeats typical of Cactus homologs in other animals, the absence of the PEST sequence and the presence of a nuclear localization signal likely prevented it from functioning as a typical I-κB inhibitor in the cytoplasm. Since homologs of genes encoding typical *Drosophila* antimicrobial proteins that are transcriptionally stimulated by the Toll pathway (e.g., *drosomycin*) were not found in ticks, we examined the expression of previously described tick immune genes. The list included: the hemocyte-specific *defensin* (*defIR*) [2], genes encoding components of the tick thioester complement-like system (*tep*, *c3*, *a2m*, and *mcr*) [21], the hemocyte-specific lectin *ixoderin A* (*ixoA*), which belongs to the Fibrinogen-related protein (FReP) family [4], and the antimicrobial gene *microplusin* [22] (S1 Table). We observed that *defIR* was significantly (> 10-fold) upregulated by KD of *cac1* and that co-silencing with *dorsal* (but not *relish*) maintained its expression at the level of dsGFP control (Figs 2 and S3). Other genes were not stimulated by KD of *cac1* or their overexpression was not restored by co-silencing with *dorsal*. These data suggest a specific role for the Cac1-Dorsal module in regulating tick innate immunity and provide evidence that the tick *defensin* (*defIR*), is a read-out gene of the tick Toll pathway.

## The tick Toll signaling pathway recognizes Gram-negative bacteria but not *Borrelia* spirochetes

In *Drosophila*, the Toll pathway recognizes Gram-positive bacteria and yeasts, while the IMD pathway reacts to Gram-negative bacteria [6]. To determine the specificity of the tick Toll pathway based on this Gram-based bacterial dichotomy, we injected representative Gram-positive and Gram-negative bacteria and yeasts into the hemolymph of tick nymphs and determined the expression of the Toll pathway read-out gene *defIR* 24 hours after treatment. In contrast to the situation in *Drosophila* [6], *defIR* expression was significantly increased in ticks injected with *Escherichia coli* (Gram-negative bacterium) and *Candida albicans* (yeast) compared to controls injected with PBS (Fig 3A). To demonstrate that this recognition is certainly specific to the Toll pathway, we repeated the experiment by injecting microbes in a *myd88* KD background. MyD88 is an intracellular signal transducer that is specific only to the Toll

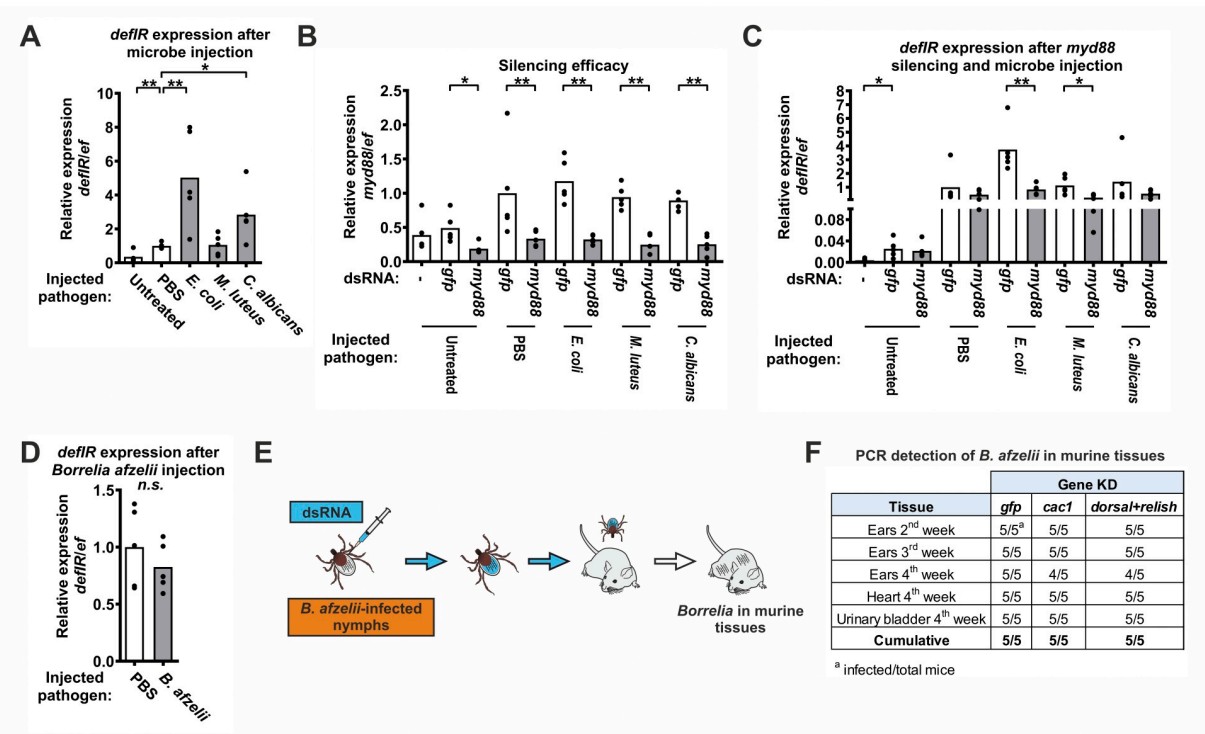

**Fig 3. The tick Toll pathway senses *E. coli* bacteria, but not *Borrelia afzelii* spirochetes.** (A) Relative expression (qRT-PCR) of *defIR* in the unfed nymphs 24 hours after injection of microbes. Each dot represents a pool of ten nymphs. (B) Efficacy of *myd88* silencing by RNAi in the unfed nymphs 14 days after injection of dsRNA and 24 hours after injection of microbes. Results represent the mean of five biological replicates. (C) Relative expression of *defIR* in the unfed nymphs 14 days after injection of dsRNA and 24 hours after injection of microbes. (D) Relative expression of *defIR* in the unfed nymphs 24 hours after injection of *B. afzelii*. (E) A scheme of *B. afzelii* transmission experiment. (F) PCR detection of *B. afzelii* in murine tissues four weeks after infestation with ten infected nymphs pre-injected with dsRNA. KD = knockdown. *P ≤ 0.05; **P ≤ 0.01; *n.s.* = not significant P ≥ 0.05.

pathway (Fig 1A). Two weeks after injection of *myd88* dsRNA, we injected the microbes into these nymphs and then measured the expression of *defIR* after 24 hours. The expression of *myd88* was significantly suppressed in all groups (Fig 3B). The expression of *defIR* in *myd88* RNAi ticks was at the level of the PBS control compared to ticks injected with dsGFP, suggesting that KD of *myd88* prevents the overexpression of *defIR* after injection of *E. coli* (Fig 3C). The *defIR* expression between *myd88* RNAi ticks and the control was also significantly reduced after injection of *Micrococcus luteus*. However, injection of *M. luteus* did not significantly upregulate *defIR* expression compared to PBS injection. We also observed a striking upregulation of *defIR* expression following injury (injection), but this response was not specific to the Toll pathway. Taken together, these results suggest that the tick Toll signaling pathway recognizes specific extracellular microbes and transmits the signal to the intracellular Cac1-Dorsal regulatory module, which then triggers expression of the effector gene *defIR*.

*Borrelia* spirochetes are atypical Gram-negative bacteria that contain neither LPS nor traditional DAP-peptidoglycans. To determine whether they are sensed by the tick Toll pathway, we injected *Borrelia afzelii*, an important causative pathogen of Lyme borreliosis in Europe transmitted by *I. ricinus*, into the hemolymph of unfed tick nymphs and measured *defIR* expression 24 hours after treatment. However, we could not detect any difference in the transcription of *defIR* compared to the PBS group (Fig 3D).

In addition, we used our established laboratory models of Lyme borreliosis [4,23,24] to test whether stimulation or blocking of the tick Toll pathway affects the (i) acquisition of *Borrelia*

spirochetes by *I. ricinus* nymphs immediately after feeding and one and two weeks after feeding on infected mice or (*ii*) transmission of spirochetes from *Borrelia*-infected nymphs to the naïve mouse. We found that neither stimulation nor blocking of the Toll pathway affected the acquisition (S4 Fig) or transmission of *Borrelia* (Fig 3E–3F). This means that *Borrelia* spirochetes are not recognized by the Toll pathway or affected by the production of Toll pathway effector molecules in the tick midgut or during their passage (through the hemolymph and salivary glands) into the host.

## Cac1-Dorsal module regulates the level of *Babesia* infection in the tick salivary glands

The Cactus-Dorsal module has been shown to regulate innate immunity in the malaria mosquito, with KD of *cactus* leading to a sharp decrease in the number of *Plasmodium* oocysts in the midgut wall [25,26]. To expand our understanding of the vector capacity of ticks as a function determined by the Cac1-Dorsal module, we developed a model involving mice infected with *Babesia microti* (a *Plasmodium*-related parasite) and *I. ricinus* nymphs. To follow the development of *Babesia* in tick tissues, we dissected the midgut and salivary glands of nymphs fed on *B. microti*-infected mice at 0, 2, 4, and 6 days post detachment (DPD) and monitored the presence of the parasite by PCR (Figs 4A and S5). *Babesia microti* was detected in the midgut and salivary glands immediately after detachment of the tick (0 DPD). We then followed the persistence of *B. microti* after the blood meal and found that mouse DNA in the midgut was degraded at 4 DPD, while *B. microti* DNA was still present, indicating active infection.

To confirm the successful invasion of the tick salivary glands by *B. microti*, we examined the tissue by confocal and electron microscopy (Fig 4B and 4C). We observed remarkable

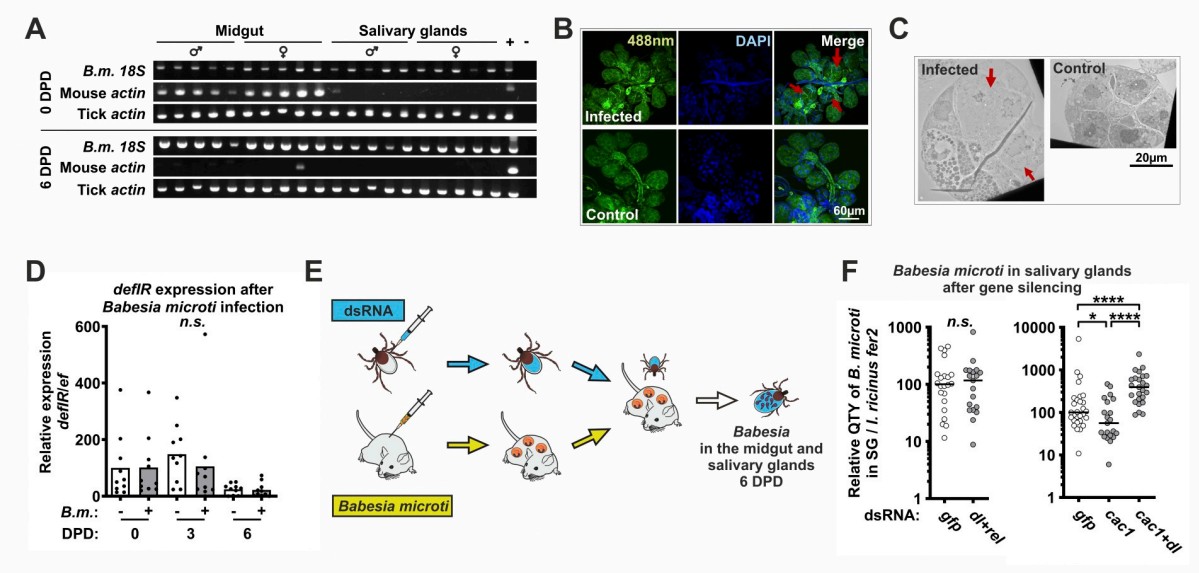

**Fig 4. The tick Toll pathway regulates acquisition of *Babesia microti* by tick nymphs.** (A) Detection of *B. microti* (*B.m.*) and mouse DNA by PCR in the nymph midgut and salivary glands at various days post detachment (DPD). Each sample represents a single tick. (B) Confocal microscopy of salivary glands 0 DPD after feeding on mice infected with *B. microti*. (C) Transmission electron microscopy of infected salivary glands 0 DPD. Red arrows indicate infected acinar cells. (D) Relative expression of *defIR* in infected nymphs (nymphs fed on *B. microti*-infected mice) 0, 3, and 6 DPD. (E) A scheme of the *B. microti* acquisition experiment. (F) Relative quantity (qRT-PCR) of *B. microti* in the salivary glands (SG) of nymphs pre-injected with dsRNA analyzed 6 DPD. The results were normalized to *I. ricinus ferritin 2* (*fer2*). rel = relish, dl = dorsal. *P ≤ 0.05; ****P ≤ 0.0001; n.s. = not significant P ≥ 0.05.

morphological changes in the organization of acinar cells compared to those of uninfected controls, suggesting the presence of parasites, as previously described [27]. *Babesia microti* appeared to have invaded type II and III acinar cells and caused marked hypertrophy. This ultrastructural change was probably related to the developing sporoblasts of the parasite, which occupy almost the entire acinus and show signs of expansion into an almost spherical, reticular mass, exhibiting a weak DAPI signal and reduced autofluorescence compared to the surrounding acinar cells. This microscopic observation, together with the above molecular evidence, suggests that *B. microti* invasion is a rapid process that begins as soon as blood is ingested. Time point 6 DPD appears to be the most suitable for quantification of *B. microti* during the acquisition experiments.

We then used this model to test whether the tick Toll pathway is stimulated during parasite infection. Tick nymphs were fed on *B. microti* infected mice and *defIR* expression was measured by qRT-PCR in whole body homogenates at 0, 3, and 6 DPD. We found that, similar to *Borrelia*, *defIR* expression was not altered in infected ticks compared to uninfected controls (Fig 4D). To assess the ability of ticks to trigger defense against *Babesia* via the Toll pathway and to reduce parasite levels during the acquisition phase, we first silenced both NF-κB transcription factors *dorsal* and *relish* to block potential stimulation of the Toll and IMD pathways. Nymphs were fed on mice infected with *B. microti*, and parasite numbers were quantified (based on DNA) in the midgut and salivary glands 6 DPD by qRT-PCR. Parasite counts in both tissues did not differ significantly from those of the dsGFP control (Figs 4E–4F and S6). We then silenced *cac1* to stimulate the expression of immune genes controlled by the Toll pathway. In doing so, we found that the burden of *B. microti* in the salivary glands decreased significantly. Simultaneous co-injection of *cac1* with *dorsal* dsRNAs abolished the *cac1* phenotype. Furthermore, KD of *cac1* together with *dorsal* not only suppressed the *cac1* phenotype but also significantly increased the number of parasites in the salivary glands compared to the dsGFP control. Finally, we silenced *defIR*, a gene that is activated by the tick Toll pathway (S7 Fig). However, we did not detect a significant difference in *Babesia* levels in both the midgut and salivary glands compared to the dsGFP control, suggesting that immune molecules other than *defIR* are involved in the regulation of *Babesia* parasites in ticks. Overall, we conclude that ticks have immune mechanisms regulated by the Toll signaling pathway, likely located in hemocytes, which can limit infection with *B. microti* to the level presumably tolerated by the tick.

## Discussion

The Toll signaling pathway is phylogenetically conserved across the animal kingdom and plays a critical role in embryonic development and immune responses to pathogenic challenges. Despite the great medical and veterinary importance of ticks, the molecular functioning of their Toll pathway has not been fully elucidated. In this work, we performed a functional assessment of the major components of the Toll pathway of the tick *I. ricinus*, an important vector of Lyme disease and other human and animal diseases. We examined the effects of RNAi-mediated gene silencing, leading to modulation (suppression or activation) of the Toll pathway, on tick embryonic development and immune responses.

By *in silico* screening, we identified the vast majority of insect homologs of the Toll pathway in *I. ricinus*, with the exception of the NF-κB transcription factor Dorsal-related immunity factor (DIF) and Gram-negative binding proteins (GNBPs) that recognize beta-1,3-glucan, a constituent of the yeast cell wall [6]. DIF, which appears to have evolved specifically in *Drosophila* and related species, is involved in adult immunity, while Dorsal plays a role in embryo development [6]. We have found that Dorsal fulfilled both developmental and immune functions in

ticks. The absence of GNBP proteins may partially explain the reported susceptibility of ticks to various entomopathogenic fungi [28]. In contrast to the well-conserved Toll pathway, several basic components of the IMD pathway (including the PGRP receptors and their adaptors IMD, FADD, and DREDD) are absent. The function of the remaining IMD pathway proteins, TAK1, IKKβ, IKKγ, Bendless, Effete, Uev1a, and Caspar, in ticks is therefore unclear. Their involvement in other signaling cascades, such as the JNK pathway [15,29], may justify their conservation. In addition, both TAK1 and the IKK complex are components of the Toll pathway in vertebrates. Recently, TAK1 has been shown to control the Toll-like receptor (TLR)-triggered antibacterial protection in shrimps (crustaceans) via activation of Dorsal [30]. Experimental studies are needed to determine the possible involvement of these molecules in the Toll pathway of ticks.

The reduction of the IMD pathway components in ticks is underscored by the presence of a gene encoding only a truncated version of Relish. The typical insect Relish consists of a REL dimerization and DNA-binding domain and the C-terminal tail of ankyrin domains that stabilize the protein in the cytoplasm. In the genome of *I. scapularis* and in the available *Ixodes* transcriptomes, we found only Relish without the ankyrin domains. A similar variant of Relish was found in the genome of the red spider mite *Tetranychus urticae* [31], and truncated Relish proteins (splice variants) have been described in mosquitoes [32], where they are involved in defense against *E. coli* [33]. However, the function and cellular location of this truncated Relish protein in ticks is not yet fully understood. Although the cleavage activation of Relish in the *I. scapularis* tick via the IMD pathway has been experimentally demonstrated [13,16,34], the molecular mechanism is not clear. Since *Ixodes* ticks lack DREDD, which normally cleaves the loop sequence between the REL and ankyrin domains of phosphorylated Relish, the above reports suggest a mechanism with unknown enzyme and substrate.

We have found marked expression of *dorsal* in both the developing ovaries of feeding ticks and laid eggs and have shown that suppression of *dorsal* expression by RNAi significantly impedes development of tick embryo. Tick eggs undergo a developmental process in the ovary in which they are attached to the luminal epithelium via a stalk of funicular cells [35]. Unlike in *Drosophila*, where maternal mRNA transfer occurs mainly through the follicle and nurse cells in the ovarioles, tick eggs in the ovary lack this surrounding cell layer. Embryogenesis of the ticks then begins in the eggs, which are laid after blood uptake and fertilization. Previously, the presence of TLR was detected in developing embryos of the tick *Rhipicephalus microti* [36] using a specific antibody. In addition, KD of *tlr* in the spider *Parasteatoda tepidariorum* resulted in abnormalities in anterior-posterior embryonic elongation, demonstrating the importance of TLR for axis formation [20]. In *Drosophila*, the Toll pathway activated by cleaved Spätzle provides dorso-ventral polarity by translocating Dorsal to the nuclei on the ventral side of the embryo [37]. These patterning processes in ticks are largely unknown, but our findings on the involvement of the Toll pathway in tick development underscore the need for further research efforts.

In post-embryonic stages of *Drosophila*, the Toll and IMD pathways can be stimulated by different microbes and cause pathway-specific expression of individual antimicrobial peptides (AMPs). The Toll pathway is stimulated by Gram-positive bacteria and yeasts and triggers the expression of, for example, *drosomycin* as a read-out gene. The IMD pathway is then stimulated by Gram-negative bacteria and triggers the expression of *diptericin*. Homologs of the representative *Drosophila* AMPs were not found in ticks. In screening for alternative AMPs in our RNAi study, we identified and genetically validated *I. ricinus defensin* (*defIR*) as an immune marker of the tick Toll pathway. The *defIR* transcript is significantly overexpressed after KD of *cac1*, which can be abolished by co-silencing of *cac1* with *dorsal*. This is not true for co-silencing of *cac1* with *relish*, confirming that the expression of *defIR* is specifically controlled by the

Cac1-Dorsal module. DefIR (JAB72371) is a typical Defensin containing a pro-peptide followed by a furin cleavage site. The mature peptide has a molecular weight of 4.9 kDa. We found that *defIR* is specifically expressed in hemocytes and according to our previous transcriptomic study, it is also the most abundant Defensin in hemocytes [2].

We observed that the expression of *defIR* was predominantly upregulated by injection of the Gram-negative bacterium *E. coli*. This upregulation can be blocked by KD of *myd88* prior to injection of the microbe. Because MyD88 is a specific adaptor only for Toll receptors and *defIR* overexpression is driven by the Cac1-Dorsal module, we have demonstrated that ticks respond to *E. coli* via the Toll pathway. This observation is in contrast to *Drosophila*, where *E. coli* is specifically sensed by IMD signaling, a pathway that is markedly reduced in ticks. However, the lack of specificity in activation of the Toll and IMD pathways by Gram-positive/negative bacteria has been previously described in the hemipteran stink bug *Plautia stali* [38] and the beetle *Tribolium castaneum* [39]. In shrimps, both the Toll and IMD pathways can recognize Gram-negative/positive bacteria and the Toll pathway can be stimulated by lipopolysaccharide (LPS) in a similar manner (direct binding of LPS to TLR) as in vertebrates [40]. This suggests that the strictly pathway-specific response to microbes observed in *Drosophila* is rather the exception and a mixture of responses is more common in other arthropods, including ticks.

Among arthropods, ticks are of particular importance because they transmit pathogens that cause serious diseases in humans and animals. In this study, we investigated whether activation of the *I. ricinus* Toll pathway regulates the internal levels of the bacterium *Borrelia afzelii* and the apicomplexan protist *Babesia microti*. *Borrelia* spirochetes are the causative agents of Lyme disease (borreliosis). The composition of the *Borrelia* envelope differs markedly from that of Gram-negative bacteria, mainly by the absence of LPS [41] and the presence of L-ornithine peptidoglycan with unusual chemical and structural properties [42]. By injecting *B. afzelii* into the hemolymph of *I. ricinus*, we tested whether the spirochetes are recognized by the tick Toll signaling pathway. Although we injected each tick with several thousand spirochetes, the expression of *defIR*, the readout gene for the Toll pathway, remained unchanged compared to the control. Consistent with this, the expression of *defIR* was not stimulated in our previously published transcriptomes of the midgut and salivary glands of *I. ricinus* nymphs infected with *B. afzelii* [17,23]. Silencing of *cac1* in *B. afzelii*-infected ticks, which led to increased expression of NF-κB effector genes, did not protect mice from infection. Also, acquisition of *Borrelia* by tick nymphs and their survival in the first two weeks were also unaffected. These results suggest that the tick does not recognize *Borrelia* via the Toll pathway and that immune molecules stimulated by this pathway are unable to kill spirochetes during acquisition or transmission, justifying *Ixodes* ticks as ideal vectors for *Borrelia* spirochetes.

Finally, we investigated the immune defense capabilities of ticks against the apicomplexan parasite *Babesia*. We chose *B. microti* and *I. ricinus* as a laboratory model due to their compatibility, as the parasite effectively infects mice and the tick nymphs feed well on rodents. The developmental cycle of *Babesia* in the tick is similar to that of *Plasmodium* in the mosquito, but with some obvious differences reflecting their life strategies (S8 Fig) [43]. Immune molecules in mosquito hemolymph, such as Thioester protein 1 (TEP1), have previously been shown to bind to *Plasmodium* ookinetes and cause their destruction. The expression of tep1 in the *Plasmodium*-infected mosquito is upregulated for several days after feeding [44] and can be stimulated by the Toll signaling pathway [25]. Moreover, KD of *cactus* by RNAi significantly reduces parasite survival in the mosquito midgut. Using our *Babesia* acquisition model, we tested whether infection of the tick with *Babesia* stimulated the Toll pathway. In contrast to the upregulation of *tep1* in infected mosquitoes, *defIR* expression in infected ticks did not differ from controls at 0, 3, and 6 days after feeding. To investigate whether stimulation of the *I. ricinus* Toll pathway affects acquisition of *Babesia*, we silenced *cac1* by RNAi and fed tick

nymphs on *Babesia*-infected mice. We expected that the resulting overproduction of Toll pathway effector immune molecules would impair the passage of *Babesia* kinetes through the hemolymph and reduce their numbers in salivary glands. Indeed, we observed a significant reduction in parasites in the salivary glands. Moreover, simultaneous silencing of *cac1+dorsal*, which completely abolished the Toll pathway response, rescued this phenotype well above the control level. Although the exact mechanism of this increase is not yet known, we hypothesize that the Toll pathway is still partially active in the dsGFP control and exerts some effect against *Babesia*, whereas its activity is completely abolished in the double KD. Our results suggest that the tick Toll pathway may be necessary to balance parasite infection rates, for example in heavily infected ticks or in infections with harmful species or strains of pathogens.

In conclusion, we demonstrated that the Toll pathway of the tick *I. ricinus* is fully functional, is important for embryonic development, and can recognize bacteria and responds to their presence by expression of antimicrobial genes. The absence of many components of the IMD pathway in ticks raises the question of whether the IMD pathway evolved exclusively in the insect lineage and complemented the Toll pathway that is widespread in arthropods. Further uncovering of all components implemented in the tick Toll pathway could provide important information about its evolution and plasticity. Our results also demonstrate that tick immunity plays a critical role in determining *Babesia* infection rates and open the door for further identification of molecules and mechanisms that define the ability of arthropods to transmit apicomplexan parasites.

## Materials and methods

### Ethics statement

All experiments were performed in accordance with the Animal Protection Act of the Czech Republic (no. 359/2012 Sb.) and Decree 501/2020 Sb. of the Ministry of Agriculture on the Protection of Experimental Animals, including relevant EU regulations, with the approval of the Ethics Committee of the Institute of Parasitology, Biology Centre, Ceske Budejovice, Czech Republic (Permit no. 25/2020).

### Biological material

Adult females of *I. ricinus* were collected in the vicinity of Ceske Budejovice (Budweis), Czech Republic, using flagging. The adults and hatched larvae were fed on guinea pigs (*Cavia porcellus*, Institutional animal facility). Molted nymphs were kept in glass boxes at 95% humidity, 24˚C, and 15/9 daylight until use. Nymphs were regularly examined for possible *Babesia* infection by PCR (*18S*, S1 Table). Six-week-old females of C3H/HeN and BALB/c laboratory mice (*Mus musculus*) were provided by Charles River Laboratories (Anlab). The experiments with *Babesia* and *Borrelia* were performed under BSL2 conditions.

### Database search and phylogenetic analysis

Tick NF-κB pathway components were searched using BioEdit 7.2.5 (NCBI Local Blast, Expectation value (E) = 0.1) with *Drosophila* and *Tribolium* protein sequences as bait in the *I. ricinus* nymph protein database generated after translation of cDNA sequences (NCBI Bioproject PRJNA657487)[17,23] by getorf (https://www.bioinformatics.nl/cgi-bin/emboss/getorf; forward and reverse sequence, nucleotide size of ORF > 90nt). The identified protein sequences were analyzed for domain structures (NCBI: CD-search) and only proteins with domain structures identical to the bait sequences and E values < 0.05 were considered reliable homologies. Signal sequences and cellular localizations were determined with DeepLoc-1.0 (https://

services.healthtech.dtu.dk/services/DeepLoc-1.0/), nuclear localization signals (NLS) with cNLS mapper (http://nls-mapper.iab.keio.ac.jp/cgi-bin/NLS_Mapper_form.cgi), and signal peptides for protein degradation (PEST) with epestfind (https://emboss.bioinformatics.nl/cgi-bin/emboss/epestfind). Phylogenetic analysis was performed in MEGA (https://www.megasoftware.net/mega4/index.php) with the settings described in the legend of Fig 1C.

### RNA interference

DNA fragments of selected genes were amplified from *I. ricinus* cDNA using gene-specific primers (S1 Table) containing ApaI and XbaI restriction sites and cloned into the pll10 vector with two T7 promoters in reverse orientation [26]. The dsRNA was synthesized using the MEGASCRIPT T7 transcription kit (Ambion) as described previously [45]. As a control, GFP dsRNA was synthesized from linearized plasmid pll6 [26] under the same conditions. The dsRNA (3 µg/µl, nymph = 32 nl, adult tick = 345 nl) was injected into the hemocoel (through the coxa III) of the unfed tick *I. ricinus* using the microinjector (Drummond). After inoculation, the ticks rested in a humid chamber for three days and were fed on mice or guinea pigs. The extent of gene silencing was verified by quantitative real-time PCR (qRT-PCR) in pools of five fully-fed nymphs or dissected tissues and compared with the dsGFP control group as described previously [23]. Feeding success, duration of feeding, and weight of individual ticks after feeding were recorded. To assess the development of the tick embryos, the tick eggs (six weeks after oviposition) were examined with a BX53 light microscope (Olympus) and classified as developed or undeveloped according to [36]. The total weight of larvae hatched from each clutch was measured after freezing the larvae (-20°C for 24 hours).

### Injection of microbes

Gram-positive *Micrococcus luteus* (CIP A270), Gram-negative *Escherichia coli* (1106), and yeast *Candida albicans* (MDM8) were cultured as previously described [46] and diluted in sterile PBS (1 x phosphate-buffered saline pH 7.3) to $OD_{600}$ = 1. *Borrelia afzelii* CB43 spirochetes were cultured in BSK-H medium as previously described [46], centrifuged at 3500 g for 10 min, washed once in PBS, and finally diluted in 0.5 ml sterile PBS (~$1 \times 10^7$ *Borrelia*/ml). *I. ricinus* nymphs were surface sterilized by immersion in 3% $H_2O_2$, 70% EtOH and distilled water and injected with 32 nl microbes as described above. The nymphs were kept in a humid chamber for 24 hours. If pretreatment with dsRNA was required prior to microbial injection, ticks were kept in a humid chamber for two weeks after dsRNA injection to reduce the immune gene expression induced by the needle injury. Then RNA was extracted from whole bodies (pool of ten nymphs per replicate).

### *Borrelia* acquisition model

Six-week-old C3H/HeN mice (Anlab) were infected by subcutaneous injection of $1 \times 10^7$ *B. afzelii* CB43. Infection of the mice was checked three weeks after injection in the ears by PCR amplification of the *flagellin* gene. Twenty dsRNA-injected (three days of rest after dsRNA injection) nymphs of *I. ricinus* were fed on each mouse (6 mice per group) until full engorgement. The number of spirochetes in the fully fed nymphs (whole bodies) was quantified by qRT-PCR (*flagellin*) immediately after feeding and one and two weeks after feeding (24 nymphs per time point, equal number of females and males) as described previously [24].

### *Borrelia* transmission model

Transmission of *B. afzelii* CB43 from the dsRNA-treated nymphs to the mice was performed as described previously [4,17,24]. Briefly, ten dsRNA-injected *B. afzelii*-infected nymphs of *I.*

*ricinus* were fed on six-week-old C3H/HeN mice (Anlab, five mice per group). Infection of the mice was checked weekly by PCR amplification of the *flagellin* gene. Four weeks after tick detachment, the mice were sacrificed and the presence of *Borrelia* in ear, bladder, and heart tissues was tested by PCR.

### *Babesia microti* maintenance

*Babesia microti* (Franca) Reichenow (PRA-99) strain Peabody mjr. (ATCC) was maintained *in-vivo* in BALB/c laboratory mice by passaging at one-week intervals. For each passage, infected mice were anesthetized (intraperitoneal injection; 100 µl of 5% Narkamon (Spofa), 2% Rometar (Spofa), and PBS at a ratio of 8:2:10) and blood was collected by cardiac puncture into citrate-phosphate-dextrose anticoagulation solution (Sigma-Aldrich; blood:solution ratio 4:1). A naïve mouse was then injected intraperitoneally with 150 µl of the blood suspension (~$5 \times 10^8$ infected red blood cells). Parasitemia in mice was monitored using blood smears (mouse tail) stained with Diff-Quik Stain (Siemens) according to the manufacturer's protocol and examined with an Olympus BX53 light microscope. Parasite stocks were prepared by mixing the infected blood solution with 30% glycerol in Alsever's Solution (Sigma-Aldrich; 10% glycerol final) and freezing the aliquots in liquid nitrogen.

### *Babesia* acquisition model in ticks

Naïve BALB/c mice were infected intraperitoneally with *B. microti* (see above for passages). One day after infection, *I. ricinus* nymphs were fed on the infected mice (20 nymphs per mouse, three mice per group, in plastic cylinders attached to the back of the mice) until they were fully engorged. In the acquisition experiments, the fully-engorged nymphs were weighed and separated into males and females with distinct weight-related sexual dimorphism [47]. Then, the midgut and salivary glands of the female nymphs were dissected six days after detachment (DPD) under a stereomicroscope (Zeiss).

### Nucleic acid extraction and cDNA synthesis

DNA and RNA were extracted using the NucleoSpin Tissue Kit (Macherey-Nagel) and the NucleoSpin RNA II Kit (Macherey-Nagel), respectively, following the manufacturer's protocol. The concentration of RNA was measured using a NanoDrop ND-1000 (Thermo Fisher Scientific). Its consistency was checked on an agarose gel. The extracted RNA was transcribed into cDNA using the Transcriptor High Fidelity cDNA Synthesis Kit (Roche) following the manufacturer's protocol (0.5 µg RNA per 20 µl reaction; anchored oligo(dT) primers) and diluted 20-fold in sterile water.

### Quantitative Real-Time PCR (qRT-PCR)

Primers (S1 Table) were designed in Primer3 (http://bioinfo.ut.ee/primer3-0.4.0/) and verified by PCR using the Fast Start Master mix (Roche). The quantification of *B. microti* (DNA) in the dissected tick organs was performed by qRT-PCR (*B. microti apical membrane antigen 1* (*ama-1*)) using a QuantStudio 6 Flex Real-Time PCR System (Applied Biosystems) and Light-Cycler 480 SYBR green I Master chemistry (Roche). The results were normalized to *I. ricinus ferritin 2* (*fer2*). The expression profiling of tick genes was done as described previously[23,48] and normalized to *I. ricinus elongation factor 1* (*ef*).

### Epifluorescence microscopy

The dissected salivary glands were fixed with 4% paraformaldehyde (Polysciences) for 10 minutes at room temperature (RT), washed 2× in PBS, and incubated in 70% ethanol for 1 hour.

Subsequently, the samples were washed 2× in PBS, stained with DAPI (4',6-diamidino-2-phenylindole dihydrochloride, final concentration 5 ng/ml in PBS) for 10 minutes and washed 2× in PBS. Finally, the slides were mounted with Fluoromount Aqueous Mounting Medium (Sigma) and examined with Olympus FW1000 confocal microscope. Images were processed using Fluoview software (FV31S- SW, version 1.7).

## Transmission electron microscopy

Dissected salivary glands were fixed overnight in a mixture of 4% paraformaldehyde (Polysciences) and 0.1% glutaraldehyde (EMS-SPI) in PBS at 4˚C. Samples were then washed 3× 10 minutes in wash buffer (4% glucose (Sigma)) and dehydrated for 15 minutes with ascending dilutions of ethanol (30%, 50%, 70%, 80%, 90%, 95%, 100%). The dehydrated samples were gradually infiltrated in LR White Resin (London Resin Company) at ratios of resin:100% ethanol 1:2, 1:1, and 2:1 (1.5 hours each, 4˚C). Finally, the samples were embedded in pure LR White Resin overnight at 4˚C, re-embedded in fresh resin, and polymerized at 50˚C for 48 hours. Samples were cut into ultrathin sections (90 nm), placed on nickel grids, and contrasted with uranyl acetate for 30 minutes and lead citrate (Sigma) for 20 minutes. The salivary glands of the ticks were examined with a transmission electron microscope JEOL JEM 1010.

## Statistical analyses

Statistical significance of differences was tested with GraphPad Prism 9.0 (GraphPad Software, CA) using the nonparametric Mann-Whitney or Kruskal-Wallis tests, and $P < 0.05$ (*), $P < 0.01$ (**), $P < 0.001$ (***), or $P < 0.0001$ (****) were considered significant.

## Supporting information

**S1 Fig. Components of the Toll pathway, but not the IMD pathway, are well conserved in ticks.** *In silico* screening of the nymphal cDNA database of *I. ricinus* (Bioproject PRJNA657487) for components of the Toll and IMD pathways. ANK = ankyrin. Details of the screening can be found in S1 Data.
(TIF)

**S2 Fig. The tick Toll pathway components regulate embryo development.** (A) Efficacy of *dorsal* silencing by RNAi in ovaries of half-fed adult females and laid eggs measured by qRT-PCR. Each dot represents a pool of five ovaries or egg clutches. (B) Duration of feeding of adult females. (C) Weight of fully-fed adult females. Results include data from three biological replicates. (D) Effect of gene silencing on development of tick embryo in eggs six weeks after egg laying. Twenty eggs from one clutch were analyzed for embryo development in each replicate. (E) A representative microscopic image of a developed and an undeveloped egg. (F) Total weight of all larvae hatched from the individual egg clutches. Each dot represents one clutch. *dl* = *dorsal*. *P ≤ 0.05; ***P ≤ 0.001; *n.s.* = not significant P ≥ 0.05.
(TIF)

**S3 Fig. Screening for genes regulated by the tick Toll pathway.** Relative expression (qRT-PCR) of selected tick immune genes in dsRNA-injected, fully-fed nymphs normalized to tick *elongation factor 1* (ef). (A) *defensin* (*defIR*, GANP01012097), (B) *α2 macroglobulin 1* (*a2m1*, MT779788), (C) *α2 macroglobulin 2* (*a2m2*, MT779789), (D) *α2 macroglobulin 3* (*a2m3*, MT779790), (E) *c3 complement component protein 1* (*c3-1*, MT779792), (F) *c3 complement component protein 2* (*c3-2*, MT779793), (G) *c3 complement component protein 3* (*c3-3*, MT779794), (H) *thioester-containing protein* (*tep*, MT779791), (I) *macroglobulin-related protein 1* (*mcr1*, MT779795), (J) *macroglobulin-related protein 2* (*mcr2*, MT779796), (K)

*microplusin* (*ricinusin*) (GBIH01001600), (L), *ixoderin A* (*ixoA*, AY341424). (M) Efficacy of silencing. The result represents the mean of five biological replicates. *cac1 = cactus1, dl = dorsal, rel = relish.* *P $\leq$ 0.05; **P $\leq$ 0.01; *n.s.* = not significant P $\geq$ 0.05.
(TIF)

**S4 Fig. The tick Toll pathway does not regulate acquisition of *Borrelia afzelii* by tick nymphs.** (A) Efficacy of gene silencing in the fully-fed nymphs (whole bodies) measured by qRT-PCR. Each dot represents a single nymph. (B) Relative expression (qRT-PCR) of *defIR* in the dsRNA-injected, fully-fed nymphs. Each dot represents a pool of five nymphs. Gene expression in the dsGFP control was set as 1. (C) Absolute number (qRT-PCR) of *B. afzelii* in the fully-fed nymphs pre-injected with dsRNA analyzed 0, 1, and 2 weeks after detachment. *cac1 = cactus1, dl = dorsal, defIR = defensin.* *P $\leq$ 0.05; **P $\leq$ 0.01; *n.s.* = not significant P $\geq$ 0.05.
(TIF)

**S5 Fig. Detection of *Babesia microti* (*B.m.*) and mouse DNA by PCR in the nymph midgut and salivary glands at various days post detachment (DPD).** Each sample represents a single tick.
(TIF)

**S6 Fig. The acquisition of *Babesia microti* by tick nymphs.** Relative quantity (qRT-PCR) of *B. microti* in the midgut of nymphs pre-injected with dsRNA analyzed 6 days post detachment (DPD). The results were normalized to *I. ricinus ferritin 2* (*fer2*). *dl = dorsal, rel = relish, cac1 = cactus1.* *n.s.* = not significant P $\geq$ 0.05.
(TIF)

**S7 Fig. The acquisition of *Babesia microti* by tick nymphs.** (A) Efficacy of *defIR* silencing in the fully-fed nymphs (whole bodies) measured by qRT-PCR. Each dot represents a pool of five nymphs. (B-C) Relative quantity (qRT-PCR) of *B. microti* in the midgut (B) and salivary glands (C) of nymphs pre-injected with dsRNA analyzed 6 DPD. The results were normalized to *I. ricinus ferritin 2* (*fer2*). *defIR = defensin, cac1 = cactus1.* *P $\leq$ 0.05; **P $\leq$ 0.01; *n.s.* = not significant P $\geq$ 0.05.
(TIF)

**S8 Fig. Schematic representation of Toll-driven tick immunity against *Babesia microti*.** (A) Development of *B. microti* in *I. ricinus*. The tick acquires the parasites by feeding (which takes several days) on host blood infected with *B. microti*. During the sexual stage of the parasite in the midgut lumen of the tick, the *Babesia* gametes release from the infected red blood cells and fuse to form motile zygotes (ookinetes). The zygotes mechanically penetrate the newly-forming peritrophic matrix and enter the midgut cells to multiply and produce kinetes. The kinetes leave the midgut cells and infect the salivary glands via the hemolymph. In the hemolymph, the kinetes are exposed to the damaging effect of immune molecules (red dots), which are mainly secreted by hemocytes. In the salivary glands, the kinetes transform into sporoblasts, which survive the molting of the tick and remain until the next blood meal, where they mature into sporozoites that invade the host red blood cells. Through the Toll pathway, the tick immune system can balance the number of *Babesia* parasites (which was shown in our experiments by silencing *cac1* and *cac1+dorsal*) and thus determine the tick vector capacity. (B) Development of *Plasmodium* in the mosquito. Ookinetes, formed by the fusion of gametes, cleave chitin in the peritrophic matrix with the help of chitinase and then traverse the midgut cells to form oocysts between the cells and basal membrane. Here the ookinetes are exposed to attack by the mosquito immune molecules (e.g., thioester protein 1 (TEP1)). After a few days,

the sporozoites break through the oocyst wall, are released into the hemolymph and migrate to the salivary gland. They accumulate in the secretory cavities and salivary gland ducts until the next feeding.
(TIF)

**S1 Table. List of primers and probes.** Restriction sites for ApaI/XbaI are underlined.
(PDF)

**S1 Data. Table of genes identified in the *in silico* screening of the nymphal cDNA database of *Ixodes ricinus* (Bioproject PRJNA657487) for the Toll and IMD pathway genes.**
(PDF)

**S2 Data. Alignment of the selected NF-κB protein sequences used in Fig 1C.**
(FAS)

**S1 Text. Descriptions of the NF-κB sequences used in S2 Data**
**.**
(PDF)

## Acknowledgments

We thank G. Loosova, J. Erhart, A. Palusova, and Z. Smejkalova for their excellent technical support and Prof. Marek Jindra for critical review of the manuscript.

## Author Contributions

**Conceptualization:** Marie Jalovecka, Laurence Malandrin, Veronika Urbanova, Radek Sima, Petr Kopacek, Ondrej Hajdusek.

**Data curation:** Marie Jalovecka, Ondrej Hajdusek.

**Formal analysis:** Marie Jalovecka, Ondrej Hajdusek.

**Investigation:** Marie Jalovecka, Laurence Malandrin, Veronika Urbanova, Sazzad Mahmood, Pavla Snebergerova, Miriama Peklanska, Veronika Pavlasova, Radek Sima, Ondrej Hajdusek.

**Methodology:** Marie Jalovecka, Sazzad Mahmood, Pavla Snebergerova, Miriama Peklanska, Veronika Pavlasova, Radek Sima, Ondrej Hajdusek.

**Project administration:** Ondrej Hajdusek.

**Supervision:** Radek Sima, Petr Kopacek, Ondrej Hajdusek.

**Validation:** Marie Jalovecka, Radek Sima, Petr Kopacek, Ondrej Hajdusek.

**Visualization:** Marie Jalovecka, Veronika Urbanova, Ondrej Hajdusek.

**Writing – original draft:** Jan Perner, Ondrej Hajdusek.

**Writing – review & editing:** Marie Jalovecka, Laurence Malandrin, Veronika Urbanova, Radek Sima, Petr Kopacek, Jan Perner, Ondrej Hajdusek.

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
