## [Decision Letter · Decision Letter 0]

13 Sep 2024

Dear Dr. Hajdusek,

Thank you very much for submitting your manuscript "Activation of the tick Toll pathway to control infection of Ixodes ricinus by the apicomplexan parasite Babesia microti" for consideration at PLOS Pathogens. As with all papers reviewed by the journal, your manuscript was reviewed by members of the editorial board and by several independent reviewers. In light of the reviews (below this email), we would like to invite the resubmission of a significantly-revised version that takes into account the reviewers' comments.

We read and discussed your appeal. While we are hesitant to changes decisions, your appeal and proposed response to reviewer comments was reasonable. As you prepare your manuscript for resubmission, please focus your attention on addressing the comments from reviewer 1 with both text and data. This may include some additional experiments and inclusion of data in modified figures and/or additional supplemental figures/data. Please also be sensitive to the wording of your response to the reviewers because these will be shared with the reviewers during the future reviews of the revised manuscript.

You will see that the reviewer's opinions differed on this manuscript. I agree with reviewer 1 who indicates that substantial efforts will be needed to revise the manuscript..

We cannot make any decision about publication until we have seen the revised manuscript and your response to the reviewers' comments. Your revised manuscript is also likely to be sent to reviewers for further evaluation.

Sincerely,

Elizabeth A McGraw, PhD

Academic Editor

PLOS Pathogens

Jeffrey Dvorin

Section Editor

PLOS Pathogens

Michael Malim

Editor-in-Chief

PLOS Pathogens

orcid.org/0000-0002-7699-2064

You will see that the reviewer's opinions differed on this manuscript. I agree with reviewer 1 who indicates that substantial efforts will be needed to revise the manuscript. I think it is best that you use this review to improve the work and then submit it elsewhere.

Reviewer's Responses to Questions

**Part I - Summary**

Reviewer #1: In this manuscript titled “Activation of the tick Toll 1 pathway to control infection of Ixodes ricinus by the apicomplexan parasite Babesia microti” Jalovecka et al conduct experiments to investigate the role of the Toll pathway in Ixodes ricinus and conclude that the pathway plays a role in tick development (based on egg development) and in restricting the infection of tick salivary glands by Babesia microti. This study highlights the role of the Toll pathway in ticks, in my opinion, and is an area of general interest to arthropod research and vector biology research. However, the results presented are not comprehensive, do not fully justify the conclusions, and largely descriptive.

In the entire manuscript the authors focus on hemocytes as key players, presumably. In the context of Borrelia and Babesia, the midguts should also be addressed. While the authors have examined Borrelia transmission in Cactus/dorsal knockout experiments, they do not address Borrelia acquisition -as the spirochetes enter the midgut to colonize the tick. Midguts are expressing these Toll components (authors show this Fig 1) – the authors have missed out on this whole important aspect of Borrelia -tick interactions. Borrelia migrates through the hemocoel during transmission, a short time when compared to its survival in the midgut over its entire time in the tick.

Despite the title’s emphasis on the role of Toll in Babesia, the whole section on Babesia is also barely developed and some of the results difficult to interpret.

The finding that Toll is involved in development is not conceptually novel, albeit important; however this section is also not well developed. Ticks have several developmental stages, and these are not examined. Egg development appears delayed -but no hard data is offered. Larvae seem to hatch with no morphological abnormalities- unclear if they looked for these phenotypes. It is not clear if these larvae were able to continue their feeding and subsequent development

The authors also jump between dorsal silencing in development, myd88 in the E.coli, M. luteus experiments and dorsal/cactus in the Babesia/borrelia experiments. I could not find any real explanation for why one component was chosen over the other. The authors show cactus 1 and Cactus 2 and do not explain why the focus was on cactus 1 , although Cactus 2 is predominantly expressed in hemocytes.

While the authors have attempted to address several aspects of tick and tick-pathogen interactions, the the reader is left with more questions than answers. Overall, the difficulties in fully reconciling the data with the conclusions presented stems from several outstanding limitations (noted below) in this report. Both development and Babesia where they observe a phenotype are only superficially addressed. The work will need significant additional experiments to gain meaningful insights.

Reviewer #2: This is an excellent manuscript by Jalovecka et al. that explores innate immune signal pathways in the tick Ixodes ricinus. It describes the conservation of the components of the Toll pathway between insects and this tick, and the connection of this pathway with tick infection by bacteria (Borrelia sp. and others) and the protozoon Babesia microti. This study uses state-of-the-art methodologies and throws light on the balance between ticks and the microorganisms they harbor, which results in an efficient vector capacity while controlling microorganism densities, minimizing in this way their deleterious effects on tick tissues.

Reviewer #3: The authors characterized components of the Toll pathway in the tick Ixodes ricinus. This done in a variety of ways. Bioinformatic analysis indicated that the Toll pathway components are well conserved, while the IMD pathway is incomplete. The authors then detected mRNA transcription of a variety of Toll components in major tick tissues and showed that the Toll components (relish1, relish2, cactus1, cactus2, defensin) were notably expressed in ovaries (development) and hemocytes (immune cells). They also screened different tick life stages and showed that these components are highly expressed in eggs and unfed larvae and to some extent in other life stages. They then selected the Toll component Dorsal and suppressed expression using RNAi. Knockdown of Dorsal affected embryonic development and confirmed the role of Toll components in development. Since Cactus and Dorsal show antagonistic interactions they investigated the impact of Cactus and Dorsal silencing on the expression of immune genes. Knockdown of cactus1 led to a significant increase in defensin expression while a range of other immune genes were not as much affected. Co-knockdown of cactus and dorsal abolished the increase in expression indicating that dorsal (and the Toll pathway) have an important effect on defensin expression. They then infected ticks with gram-negative or gram-positive bacteria or yeast and showed that only gram-negative bacteria elicit upregulation of defensin expression suggesting that the Toll pathway is activated when exposed to Gram-negative bacteria. This is of interest, since in insects gram-negative responses is mediated by the IMD pathway and not the Toll pathway. Conversely, no Toll response was observed when ticks were infected with Borrelia afzelii. Knockdown of cactus1 and co-knockdown of dorsal and relish did not affect Borrelia transmission. The effect of the Toll pathway was further investigated for Babesia microti infection and transmission and it was shown that the Toll pathway has no effect. As such, two natural pathogens transmitted by I. ricinus does not stimulate the Toll pathway.

The study was well executed and the experimental approach well described. The results that the Toll pathway (dorsal) impacts development is convincing and also that this pathway leads to expression of defensin, one of the main anti-microbial defense mechanisms in ticks. It highlights the conservation of the Toll pathway compared to the IMD pathway and the functionality of the Toll pathway. This is definitely an important advance in the understanding of tick immunity. Nice models to study Borrelia afzelii and Babesia microti is also presented and importantly that the Toll pathway is not stimulated by these natural pathogens. The study do raise numerous questions that can be addressed in future research such as whether these pathogens elicit any immune response at all (a case of remarkable co-adaptation) and or which response would then be activated. I believe that trying to answer these in the current study would expand the study too much. Very few issues were raised and I relegate these to minor revision.

**Part II – Major Issues: Key Experiments Required for Acceptance**

Reviewer #1: Cact1 is expressed in ovaries and hemocytes; - makes sense that it is involved in development. Cac2 is expressed predominantly in hemocytes (line 133). Yet it is only Cact1 that is targeted in all their studies. I could not find an explanation for why only Cact1 was being targeted. What would be the phenotype in the context of Cact2 targeting? In line 168, they emphasize the cact1-dorsal module in the Toll pathway but given that cact2 is expressed in hemocytes -showing no data on cact2 leaves a major gap in the conclusions presented. This would shed further insights into this pathway.

Also, only dorsal is targeted in the development phenotype experiments. Would be important to know the impact of Cactus 1 targeting in this, given that this is also predominant in the ovaries. In the development part of the manuscript, Fig 2 notes that embryo development was dramatically reduced in Fig 2B (Fig 2C -not clear what was being observed as legs? a better image is requested), however this is not consistent with the larval hatching impact in 2D. Was egg laying delayed in anyway? Was egg development monitored over time? This leg development parameter as a score of development could be monitored over time to score for the developmental delay. Without this information, the inconsistency in the strength of the impact between Fig 2B and 2D is confusing.

In injection experiments with E. coli or M. luteus, the authors knockdown the expression of myd88 and wait for 2 weeks before injecting with the pathogens. Unclear why the wait was required. Also not explained in the Methods section.

In the E. coli, M. luteus experiments -did they check for any readouts in the IMD pathway? AMP such as Ctenidin was shown to be activated by the IMD pathway (ONeal et al, 2023).

Upon knockdown of Myd88 and injection of E. coli, the authors do not monitor tick survival, and this further begs the question as to whether Toll pathway restricts such bacteria in nature. This also raises another aspect of this study – using bacteria such as E. coli and M. luteus may not really be physiologically relevant.

Further, the authors seem to make abrupt jumps from one component of the pathway to the other. Why did they choose to knockdown myd88 in the E. coli/M. luteus experiments, but switched to dorsal/cactus in the Borrelia /Babesia experiments?

In the Babesia experiments, it is unclear what the purpose of the IFA and EM pictures are for; perhaps these images are merely to emphasize the model? Babesia experiments just leave the conclusion to much “speculation” on the part of the reader. While Babesia infection does not alter the DeflR transcript levels (just as in the case of Borrelia), knockdown of cactus provides decreased burden of Babesia in the salivary glands. Since the cacust1-dorsal module did not impact any of the TEP-like components in tick hemocytes, are the authors suggesting that Babesia parasites are being killed by DeflR? This is very speculative. If defensin like AMPs can be shown to kill Babesia in vitro-then the observations can be more mechanistically interpretable, but no such plausible mechanism is provided.

Lines 384-390 – very difficult to follow the rationale. Complicated further by Suppl Fig S1 that blood feeding induced DeflR expression and was not further increased upon Cac-1 silencing. The authors suggest that this is also consistent with the observation that dorsal +relish knockdown had no impact on Babesia, but dorsal +cactus did? It makes the arguments convoluted with no additional data to validate such speculations. If Dorsal-the transcription factor, induces the transcription of effector molecules dorsal +relish should also result in increased Babesia burden ?

For Babesia acquisition, midguts and salivary glands were dissected only from female (based on engorgement weights) nymphs, why not the male nymphs? Also, why only SG data are shown? How about the midgut burden?

Reviewer #2: Line 452: Please include information on how infections by these bacteria were verified. Also, were nymphs tested for the presence of these bacteria before the experimental infections were carried out? This should be something to take into account since previous exposure might affect their immune responses.

Reviewer #3: None

**Part III – Minor Issues: Editorial and Data Presentation Modifications**

Reviewer #1: Fig 2C – was hard to see which treatment shows significant increase/decrease in DeflR transcripts – the asterisks appear to be on top of the M. luteus group (presumably it should be on the E. coli group).

There are two dorsal like proteins in Drosophila: DIF and Dorsal. One involved in development and one in immune response. I. scapularis has only one – and lacks the DIF (Dorsal-like immune factor) so also I. ricinus. The authors could provide some insights on this in the Discussion.

The quantitative PCR results were normalized to ferritin for Babesia? For all other genes, results were normalized to elongation factor. An explanation for this would be helpful.

Reviewer #2: Lines 409 and 444: add references to the methodology employed for PCR and qRT-PCR.

Line 478: Use the correct scientific notation, in this case: 5 x 10 to the 8.

References: Use italics for scientific names.

Reviewer #3: Line 31: Can vector competence be determined by the ability of pathogens to evade recognition? Would vector competence not focus on the vectors ability to acquire, maintain and transmit a pathogen to a host.This latter definition focus on the vector, but the former imply that the pathogen determine vector competence. What if parasite and vector co-evolved that the pathogen is not destroyed by the immune system. This is not evasion, but co-adaptation would increase vector competence. Perhaps rephrase the sentence?

L223, 231: Write out genus name when starting a sentence.

References

Check that species names are italicized and that the references follow the journal format.

PLOS authors have the option to publish the peer review history of their article (what does this mean?). If published, this will include your full peer review and any attached files.

Reviewer #1: No

Reviewer #2: No

Reviewer #3: No
---

## [Editor Report · Decision Letter 1]

11 Nov 2024

Dear Dr. Hajdusek,

We are pleased to inform you that your manuscript 'Activation of the tick Toll pathway to control infection of Ixodes ricinus by the apicomplexan parasite Babesia microti' has been provisionally accepted for publication in PLOS Pathogens.

Best regards,

Elizabeth A McGraw, PhD

Academic Editor

PLOS Pathogens

Jeffrey Dvorin

Section Editor

PLOS Pathogens

Michael Malim

Editor-in-Chief

PLOS Pathogens

orcid.org/0000-0002-7699-2064
---

## [Editor Report · Acceptance letter]

27 Nov 2024

Dear Dr. Hajdusek,

We are delighted to inform you that your manuscript, "Activation of the tick Toll pathway to control infection of Ixodes ricinus by the apicomplexan parasite Babesia microti," has been formally accepted for publication in PLOS Pathogens.

Best regards,

Michael Malim

Editor-in-Chief

PLOS Pathogens

orcid.org/0000-0002-7699-2064